# Novel Biomolecules in the Pathogenesis of Gestational Diabetes Mellitus 2.0

**DOI:** 10.3390/ijms23084364

**Published:** 2022-04-14

**Authors:** Monika Ruszała, Aleksandra Pilszyk, Magdalena Niebrzydowska, Żaneta Kimber-Trojnar, Marcin Trojnar, Bożena Leszczyńska-Gorzelak

**Affiliations:** 1Department of Obstetrics and Perinatology, Medical University of Lublin, 20-090 Lublin, Poland; monika.ruszala@wp.pl (M.R.); apilszyk@gmail.com (A.P.); mniebrzydowska7@gmail.com (M.N.); b.leszczynska@umlub.pl (B.L.-G.); 2Department of Internal Diseases, Medical University of Lublin, 20-059 Lublin, Poland; marcin.trojnar@umlub.pl

**Keywords:** gestational diabetes mellitus, biomolecules, CMPF, ANGPTL-8, nesfatin-1, afamin, adropin, fetuin-A, zonulin, SFRPs

## Abstract

Gestational diabetes mellitus (GDM) has become a major public health problem and one of the most discussed issues in modern obstetrics. GDM is associated with serious adverse perinatal outcomes and long-term health consequences for both the mother and child. Currently, the importance and purposefulness of finding a biopredictor that will enable the identification of women with an increased risk of developing GDM as early as the beginning of pregnancy are highly emphasized. Both “older” molecules, such as adiponectin and leptin, and “newer” adipokines, including fatty acid-binding protein 4 (FABP4), have proven to be of pathophysiological importance in GDM. Therefore, in our previous review, we presented 13 novel biomolecules, i.e., galectins, growth differentiation factor-15, chemerin, omentin-1, osteocalcin, resistin, visfatin, vaspin, irisin, apelin, FABP4, fibroblast growth factor 21, and lipocalin-2. The purpose of this review is to present the potential and importance of another nine lesser known molecules in the pathogenesis of GDM, i.e., 3-carboxy-4-methyl-5-propyl-2-furanpropanoic acid (CMPF), angiopoietin-like protein-8 (ANGPTL-8), nesfatin-1, afamin, adropin, fetuin-A, zonulin, secreted frizzled-related proteins (SFRPs), and amylin. It seems that two of them, fetuin-A and zonulin in high serum levels, may be applied as biopredictors of GDM.

## 1. Introduction

Gestational diabetes mellitus (GDM) has become a major public health problem and one of the most discussed issues in modern obstetrics, given that it is one of the most common metabolic disorders in obstetrics [1,2,3]. The scientific literature defines GDM as a state of hyperglycemia developing in pregnancy as a result of insulin resistance or reduced insulin production, which resolve following delivery [4,5,6,7]. Given the various diagnostic criteria and ethnicity, in many countries the epidemiology of GDM still remains unknown [8,9]. It is estimated that the prevalence of GDM has increased by more than 35% in the last few decades and is still growing, which is rather alarming [10]. The risk factors for GDM include pre-pregnancy overweight and obesity, advanced maternal age, excessive gestational weight gain (EGWG), family history of type 2 diabetes mellitus (T2DM), GDM during previous pregnancies, previously giving birth to a baby with a birth weight greater than 4000 g, multiparity, and polycystic ovary syndrome (PCOS) [11,12].

A pregnant woman’s metabolism determines both the mother’s own health and the health of her child [13,14,15]. Excessive intake of carbohydrates and lipids, reduced consumption of vegetables and fruit, reduced physical activity, and genetic predisposition may lead to the development of hyperglycemia. Peripheral insulin resistance, which gets worse with each passing week of pregnancy, involves a physiological maternal adaptation process that makes it possible for the mother to comply with the increasing energy demand of a rapidly developing fetus [16]. It is observed that insulin resistance increases considerably in pregnant women with pre-pregnancy overweight and obesity in comparison to those with pre-pregnancy normal weight. This triggers immune and inflammatory responses in the white adipose tissue, which in turn leads to the development of a low-grade systemic chronic inflammation known as metabolic inflammation [17]. It is common knowledge that an increased inflammatory response associated with excessive body fat is a key factor in reducing the action of insulin [16,18,19,20]. Furthermore, it also reduces the β-cell compensatory response, which promotes the development of GDM [21]. In the vicious cycle of GDM, the production of various pro-inflammatory cytokines increases, whereas the expression of anti-inflammatory biomolecules is inhibited.

Elevated glucose levels stimulate the pancreas to release insulin, and the tissues and cells developing insulin resistance stimulate the pancreatic β cells to produce more insulin, thereby impairing their function. Exposure to maternal hyperglycemia may impede fetal development and cause many adverse pregnancy outcomes, such as a high risk of premature birth, macrosomia, neonatal respiratory complications, postnatal hypoglycemia, and hypoxia [22,23]. Also, exposing the fetus to persistent hyperglycemia induces ‘glycemic memory’ in the fetus, which may contribute to epigenetic changes, i.e., DNA methylation disturbances [24,25]. Furthermore, a hypoxic memory can affect the kidneys and cause an acute kidney injury [26]. Hyperglycemia may also impair the autonomic nervous system (ANS). Changes in the fetal ANS are manifested by high blood pressure, tachycardia, and respiratory complications [27]. Extended hypoxia enhances angiogenesis, endothelial dysfunction, cell proliferation, and inflammation [28,29,30]. GDM can also have long-term consequences for the offspring, such as an increased risk of obesity, T2DM, and cardiovascular diseases (CVDs) in their adult life [28]. On the other hand, women with a history of GDM are also at a higher risk of maternal programming for the development of diseases of civilization later in their lives, with T2DM being the most prevalent among them. Women who have had GDM are at a 35–60% higher risk of developing T2DM in the next 10–20 years [31].

The COVID-19 pandemic has changed the world situation considerably. Social distancing, less physical activity, and changes in lifestyle and eating habits have contributed to increased weight gain [32,33]. The excessive energy balance of the body promotes the accumulation of adipose tissue and, consequently, the overexpression of adipokines, which possess pro-thrombotic, pro-inflammatory, pro-atherosclerotic, and pro-diabetic properties. Even though there is a clear-cut relationship between obesity and diabetes [17,34], there is still no answer to the question of why this disease does not develop in all obese people and how its development can be predicted.

Although GDM cannot be completely prevented, its early diagnosis and prompt management significantly improve both the development of the fetus and the course of pregnancy, delivery, and the postpartum period. There is a strong need to focus on disease screening. The greater the number of available diagnostic methods, the more likely it is to detect the disease at its early stage. Currently, the scientific literature emphasizes the importance of finding biopredictors that will allow for the identification of women with an increased risk of developing GDM at the beginning of pregnancy. Many adipokines have proven to be of pathophysiological importance in GDM. These include both “older” molecules, such as adiponectin and leptin, as well as “newer” adipokines, including fatty acid binding protein 4 (FABP4) [35,36,37,38,39]. In our previous review, we presented 13 new biomolecules, i.e., galectin, growth differentiation factor-15 (GDF-15), chemerin, omentin-1, osteocalcin, resistin, visfatin, vaspin, irisin, apelin, FABP4, fibroblast growth factor 21 (FGF21), and lipocalin-2 [40]. The purpose of this review is to present the potential and importance of another nine new molecules in the pathogenesis of GDM that are less known but appear to be no less interesting. Therefore, in this paper we would like to describe the following substances: 3-carboxy-4-methyl-5-propyl-2-furanpropanoic acid (CMPF), angiopoietin-like protein-8 (ANGPTL-8), nesfatin-1, afamin, adropin, fetuin-A, zonulin, secreted frizzled-related proteins (SFRPs), and amylin. We are discussing their potential in the detection of GDM, which might play a significant role in both maternal and fetal health outcomes.

## 2. Biomolecules

### 2.1. CMPF

The furan fatty acid metabolite, CMPF—a fish oil metabolite—is an endogenous uremic toxin involved in the glycolipid processes and islet β-cell dysfunction that enters the β-cell through the human organic anion transporter 3 (OAT3) [41,42]. CMPF is also associated with thyroid dysfunction. It has the ability to cross the blood-brain barrier, thereby contributing to various neurological abnormalities. Abnormal concentrations of CMPF were detected in patients with colorectal adenoma [43]. CMPF was first detected in human urine in 1979. CMPFs are coupled with triglycerides, phospholipids, and cholesterol esters [44]. They demonstrate high protein-binding ratios (above 95%) and are found in green plants, champignons, vegetable oils, fish, or algae [45,46]. In humans, CMPF cannot be synthesized de novo. A metabolite of the furan fatty acid, dicarboxylic acid, is excreted in urine, and it may be detected in serum, feces, and urine. The CMPF blood levels differ between males and females, and they are significantly elevated in the former. Many factors may affect the CMPF levels in serum and they include: inter alia, ethnic background, and dietary and/or metabolic components.

The role of CMPF in the etiology of GDM is still unclear [47,48]. Some studies have shown that the level of CMPF may be increased in patients with T2DM and GDM [49,50]. It might also be accumulated in patients with kidney diseases [51]. Since the early 1990s, it has been suggested that CMPFs contribute to renal tubular damage by interacting with the superoxide anion and peroxy radicals. It is worth adding that an elevated serum concentration of CMPF is not associated with mortality and cardiovascular morbidity. Ji et al. conducted a study that revealed significantly higher serum CMPF levels in the investigated GDM group of patients in comparison to healthy controls [52]. Moreover, there was a positive correlation between an elevated CMPF serum level and glycated hemoglobin (HbA1c), fasting plasma glucose, one-hour plasma glucose, and two-hour plasma glucose in an oral glucose tolerance test (OGTT). Additionally, CMPF manifested no correlation between total cholesterol, triglycerides, high-density lipoprotein (HDL), and low-density lipoprotein (LDL), which were increased in the GDM group. In another study conducted by Prentice et al., patients with impaired glucose tolerance and GDM still had elevated CMPF serum levels, even one year after delivery [49]. Liu et al. underlined the fact that patients with an upregulated furan fatty acid metabolite may develop T2DM within the next 5 years [50]. The same study also reports that those women who developed T2DM had had significantly higher levels of CMPF 4 years previously, while pre-diabetics maintained elevated but stable CMPF levels [50].

In the scientific literature there are many theories regarding the CMPF working action mechanisms. It is assumed that this molecule may contribute to the development of diabetes by reducing insulin secretion and synthesis. Moreover, it stimulates oxidative stress, acts on the mitochondrial functions, and dysregulates glucose-induced adenosine triphosphate (ATP) accumulation [49]. In addition, CMPF increases β-oxidation, reduces lipogenic gene expression, and improves the clinical condition of patients with steatosis. CMPF is also indicative of anti-inflammatory effects. It has been observed that an extract obtained from green lipped mussels, which contain furan fatty acids, alleviated symptoms in patients with rheumatoid arthritis. The mechanism depends on a decrease in the level of interleukin (IL) 1β, prostaglandin (PGE2), and tumor necrosis factor α (TNF-α) [53]. Moreover, CMPF downregulates the toll-like receptor 4 (TLR-4) signaling pathway. These results were in agreement with the findings of Lankinen et al., who demonstrated a correlation between increased CMPF serum levels and reduced two-hour insulin levels in the OGTT test [54]. Acting upon these results, Retnakaran et al. surmised that the furan fatty acid metabolite might be a potential marker for the islet β-cell dysfunction and hyperglycemia in pregnant women diagnosed with GDM [55]. Nevertheless, the authors also revealed that CMPF was not a significant determinant of the postpartum whole-body insulin sensitivity index (Matsuda index) or homeostatic model assessment for insulin resistance (HOMA-IR). Serum CMPF concentrations did not differ between women with postpartum normal glucose tolerance (NGT) and those with postpartum prediabetes/diabetes [56].

Prentice et al. indicated that the effect of CMPF on the islet β-cells can be reversed with benzylpenicillin or probenecid via OAT3 (Slc22a8) [49]. It is supposed that some factors such as race, the onset of diabetes, age, or dietary habits may affect the total CMPF serum level. In the research conducted by Zheng et al., patients with T2DM had lower CMPF serum levels in comparison to healthy controls [57]. Florian et al. did not obtain significant values of CMPF in patients with GDM either [41]. The research conducted by Savolainen et al. revealed an inverse correlation between CMPF and the risk of T2DM development in the studied Swedish women [58]. Different results were obtained by Fardipour et al. in 2020 because serum CMPF concentrations were elevated in Iranian pre-diabetic patients [59]. These authors concluded that CMPF may be used as a biomarker for the diagnosis of pre-diabetes [59].

CMPF seems to be a promising marker that could be used to assess the β cell function and signal the risk of hyperglycemia [60]. It could be helpful in improving maternal–fetal prognosis and in the prevention of cardiovascular and neural complications [41]. However, these beneficial results are not confirmed by scientific studies yet.

### 2.2. ANGPTL-8

The structure of ANGPTLs is similar to angiopoietins [61]. They are secreted by the adipose tissue, liver, and vascular and hematopoietic systems, and their main functions include the regulation of glucose homeostasis, lipid metabolism, inflammation, and angiogenesis [62,63,64]. ANGPTL-8, otherwise known as betatrophin, TD26, lipasin, or C19orf80, is secreted mainly by the liver and adipose tissue, and its role seems to be significant in lipid metabolism and the maintenance of glucose homeostasis. It appears that the disruption of its function may contribute to the development of GDM during pregnancy [64,65].

It has been observed that the expression of ANGPTL-8 is stimulated by insulin, food intake, and cold exposure, while it is inhibited by starvation [63,66]. ANGPTL-8 seems to play a supporting role in stimulating proliferation and increasing the pancreatic β-cell mass as well as improving glucose tolerance in insulin resistance, presumably by increasing insulin secretion [64,67]. However, not all researchers see eye to eye about this. Jiao et al. were able to document pancreatic β-cell replication by stimulating ANGPTL-8 in mice, however, they failed to increase human β-cell DNA replication in the transplanted setting [68].

There are reports that ANGPTL-8 levels are dependent on lipid and carbohydrate disorders, which appear to be related to its inhibition of the lipoprotein lipase enzyme activity [63,69]. The ANGPTL-8 gene contains a carbohydrate responsive element that is activated by high glucose and lipid levels [61]. It also appears to be associated with the development of type 1 diabetes [61], T2DM [70], hypertension, and metabolic syndrome [69].

It has been observed that the serum level of ANGPTL-8 in healthy pregnant women is higher than in non-pregnant ones, and it decreases rapidly after delivery [69]. Moreover, ANGPTL-8 concentrations are higher in the cord blood than in the maternal serum. This may suggest its important role in maintaining pregnancy and influencing fetal development [71]. At the same time, many researchers have observed a significant increase in the levels of this adipokine in pregnant women with GDM compared to healthy controls [61,62,63,64,66,69,72]. Some researchers believe that high levels of ANGPTL-8 during gestation may be responsible for the pancreatic β-cell proliferation and assessment of its levels can significantly contribute to the early diagnosis and prediction of GDM development [71].

Abdeltawab et al. [61] conducted a scientific study in which they compared the ANGPTL-8 levels in 109 pregnant women with GDM diagnosed between 24 and 28 weeks of gestation and in 103 healthy controls. They observed a significant increase in the levels of this adipokine in the women suffering from GDM in comparison to healthy controls. However, the authors emphasize that further studies of ANGPTL-8 levels during the first trimester of pregnancy are required to assess whether its levels are increased before the 24th week of gestation. This would help to evaluate the usefulness of this adipokine as a predictive marker of GDM development [61].

In their study, Huang et al. [69] evaluated ANGPTL-8 concentrations in 474 pregnant women. Among them, 88 developed GDM. In their report, ANGPTL-8 levels were increased in the women with GDM compared to healthy controls. Moreover, they observed that ANGPTL-8 concentrations were able to predict the development of GDM with a higher probability than body mass index (BMI) during early gestation, making this adipokine a potential predictive marker of the development of GDM independently of other markers, such as maternal age and BMI [69]. A study conducted by Seyhanli et al. brought similar results [64].

Pan et al. [72], in addition to investigating the relevance of ANGPTL-8 as a potential diagnostic marker of GDM, also focused on assessing its predictive value in the development of T2DM after pregnancy. They observed that ANGPTL-8 levels were significantly elevated in patients with GDM and showed an association with the disease that was independent of other parameters, such as BMI or HOMA-IR values. They also suggested that increased ANGPTL-8 levels during pregnancy were linked to an increased risk of developing T2DM after delivery. However, they admitted that their study had several limitations, including the fact that the BMI as well as age of the pregnant patients with GDM were higher in comparison to healthy controls. These limitations did not allow them to clearly state whether ANGPTL-8 would perform well as a reliable marker of GDM occurrence [72].

Yang et al. [73] compared ANGPTL-8 levels in 40 women with GDM and 37 healthy controls. In contrast to other researchers, they observed significantly lower concentrations of ANGPTL-8 in the sera of GDM patients compared to healthy controls. They also demonstrated a negative correlation between ANGPTL-8 levels and HOMA-IR, suggesting that decreased levels of ANGPTL-8 may be caused by insulin resistance. ANGPTL-8 concentrations in the cord blood were also significantly reduced in GDM patients but were found to be higher than in the maternal serum in both investigated groups. Interestingly, they also showed a negative correlation between ANGPTL-8 concentrations in the cord blood and higher neonatal weights of infants born to GDM mothers [73].

### 2.3. Nesfatin-1

Nesfatin-1, derived from the precursor protein nucleobindin-2 (NUCB2), consists of an 82-amino acid peptide [74]. It is secreted by the peripheral tissues, such as the pancreas, duodenum, and adipose tissue, as well as the peripheral and central nervous system (arcuate, paraventricular nuclei, and nucleus of the solitary tract) [75,76].

One of the key functions of nesfatin-1 is the regulation of carbohydrate metabolism [77]. It stimulates the pre-proinsulin mRNA expression and increases glucose-induced insulin release. Nesfatin-1 also inhibits glucagon secretion [78]. The NUCB2/nesfatin-1 cells have been observed to be localized on the pancreatic β-islets in mice and rats [79]. It is believed that nesfatin-1 is released from the β-islets of the pancreas as a response to the exposure of these cells to glucose [80]. When intravenously injected to hyperglycemic animals, nesfatin-1 had an antihyperglycemic effect [81]. However, the oral glucose tolerance test (OGTT) in healthy patients did not affect NUCB2/nesfatin-1 levels, which may suggest a local activity of nesfatin-1 around the β-islets of the pancreas rather than an endocrine effect [78].

Nesfatin-1 is also secreted by the hypothalamus, and thus affects the regulation of hunger and satiety, which secondarily contributes to body weight regulation [74]. An increase in the nesfatin-1 concentration has been observed after food consumption [77]. It is believed that nesfatin-1 may decrease food intake by reducing appetite and inducing satiety [82]. On the other hand, it is likely that reduced nesfatin-1 levels may be associated with hyperphagia in T2DM [75]. These findings suggest that nesfatin-1, via its inter alia anti-hyperglycemic and anorexigenic effects, may significantly affect metabolic regulation [81]. Thus, nesfatin-1 may serve as an important protective factor in the development of GDM [19].

Some researchers report decreased nesfatin-1 levels in patients with T2DM [83,84] and PCOS [77,84], and both of these diseases are known to be associated with insulin resistance and obesity. Therefore, it is suggested that nesfatin-1 secretion may be inhibited by insulin resistance, hyperglycemia, hyperinsulinemia, or obesity, which could explain why nesfatin-1 levels are decreased in these conditions [75,77]. It seems that reduced serum nesfatin-1 levels in pregnant women could impair insulin release in patients with GDM [79]. Most researchers evaluating the association between nesfatin-1 levels and the development of GDM in pregnant women report a significant decrease in nesfatin-1 levels during GDM [75,77,79,83,84,85].

Other researchers are of the opinion that nesfatin-1 concentrations may increase proportionally with obesity in T2DM and in non-diabetic patients. It is also suggested that obesity is the most significant risk factor for the development of GDM and the percentage of body fat is the main determinant of nesfatin-1 levels [80]. However, its effect on the development of GDM is uncertain. According to some studies, nesfatin-1 may lower blood glucose levels, which could explain its increase in patients with hyperglycemia during pregnancy [80].

According to a study by Ademoglu et al. [83], who compared nesfatin-1 levels in 40 GDM patients and 30 healthy controls, nesfatin-1 concentrations were significantly reduced in GDM patients in comparison to healthy women. In their study, they found a positive correlation between nesfatin-1 levels and gestational age. However, no association was observed between nestfatin-1 levels and BMI, HOMA-IR, or fasting glucose levels [83]. Kucukler et al. [75] also observed that patients with GDM had decreased nesfatin-1 levels in comparison to healthy controls. However, they found a negative correlation between nesfatin-1 concentrations and body weight, BMI, and fasting glucose [75].

Another study, conducted by Aydin [85], investigated the presence of nesfatin-1 in breast milk. This author found reduced nesfatin-1 concentrations in women with GDM in comparison to healthy controls. Besides, this researcher expected to show a correlation between nesfatin-1 levels in lactating women and their BMI due to their increase in body weight during pregnancy, but no such correlation was observed [85].

The study by Mierzyński et al. [77] compared nesfatin-1 levels in 153 patients suffering from GDM with nesfatin-1 levels in 84 healthy controls, and their study revealed significantly reduced nesfatin-1 levels in women with GDM in comparison to healthy controls. The concentration of nesfatin-1 in their study positively correlated with the pre-pregnancy BMI and BMI at the time of blood sampling in the group of patients with GDM and the control group. There was also a correlation between nesfatin-1 levels and glucose levels in OGTT. Mierzyński et al. also found that a 1 ng/mL increase in nesfatin-1 decreased the risk of GDM by 9.85%, respectively (CI 95%). Another observation they made was a positive correlation between nesfatin-1 levels and gestational age in the healthy group. This could suggest an increase in nesfatin-1 levels as a mechanism preventing the development of GDM during the duration of pregnancy. Decreased levels of nesfatin-1 in GDM patients could explain its involvement in the pathogenesis of GDM [77].

However, not all researchers came to similar conclusions. A meta-analysis carried out by Sun et al. [76] showed no significant difference in nesfatin-1 levels between GDM patients and healthy controls. When a detailed analysis was conducted in subgroups, it was revealed that the studied Caucasian women with GDM had lower nesfatin-1 levels, however, no similar correlation was observed in the studied Asian patients. This suggests a potential association of the predictive value of nesfatin-1 in GDM with patients’ ethnicity [76].

Another study, by Zhang et al. [80], was conducted to determine whether the levels of nesfatin-1 and its expression in the subcutaneous adipose tissue (SAT) are altered in GDM patients. Moreover, they also examined its association with biochemical tests such as glucose, lipid metabolism, and IR. As a result, they observed a significant increase in the levels of nesfatin-1 in the maternal plasma and cord blood as well as an increase in nesfatin-1 expression in SAT in those women who developed GDM in comparison to healthy controls. Zhang et al. also found a correlation between the plasma nesfatin-1 levels in the mother and her pre-pregnancy BMI, pre-partum BMI, and HOMA-IR. However, they did not find any correlation between the level of nesfatin-1 in the maternal plasma and nesfatin-1 expression in SAT or its level in the cord blood [80].

### 2.4. Afamin

Afamin is a glycoprotein that belongs to the albumin family, as do α-fetoprotein and albumin [86,87]. It is mainly produced by the liver [86] and peripheral tissues, such as the placenta [88].

Afamin is able to bind vitamin E in the extravascular fluids, and, due to hormonal changes, its levels increase during the duration of pregnancy [89,90]. Vitamin E acts as an antioxidant in the pancreatic cells, and animal studies have shown its anti-apoptotic effects against them [87,91].

Owing to its anti-apoptotic and antioxidant qualities [90,91], afamin is believed to contribute to the effects of oxidative stress [26,30]. It has been observed that an increase in oxidative stress and the occurrence of related conditions (metabolic syndrome, T2DM, insulin resistance, obesity) [89,91] correlate with elevating the serum afamin concentrations [91]. An imbalance between oxidants and antioxidants appears to be related to the development of diabetic complications and the development of GDM [91].

Given its properties, afamin is thought to be useful as an early marker of carbohydrate and lipid disorders during pregnancy [88]. Elevated levels of afamin seem to be related to the conditions that increase the risk of metabolic syndrome, such as dyslipidemia, hypertension, impaired carbohydrate metabolism, and obesity [87,90]. A study conducted in mice revealed a strong correlation between the increased afamin levels and development of metabolic syndrome and its components, i.e., hyperlipidemia, hyperglycemia, and increased body weight [92].

The role of afamin in the pathogeneses of both T2DM and GDM is suggested by its association with the occurrence of insulin resistance [86]. Increased insulin resistance at the beginning of pregnancy significantly correlates with the development of GDM later during gestation [87]. In vitro studies have shown a direct association between afamin and glucose concentrations [86]. The levels of afamin are independent of fasting status, age, sex, and they increase linearly by approximately 2-fold during an uncomplicated pregnancy [87].

Furthermore, it seems that afamin may be connected to other pathologies associated with pregnancy. There is some evidence of a correlation between increased afamin concentrations in the first trimester of pregnancy and the occurrence of pre-eclampsia [93]. 

Köninger et al. [87] conducted a study in which they made an attempt to assess the predictive and diagnostic value of afamin levels in the development of GDM in the first and second trimester of pregnancy. During the first trimester, of the 110 patients screened, 59 developed GDM later in gestation; the remaining 51 who did not develop GDM served as controls. They also analyzed 105 mid-trimester samples, of which 29 developed GDM. A total of 215 samples from pregnant women were analyzed in the study. Those women who had higher levels of afamin in the first trimester were more likely to develop GDM later during pregnancy. There was also a correlation between afamin concentrations and the treatment of GDM, i.e., patients treated with insulin had higher levels of afamin than those who were on a diet treatment. Therefore, they were unable to determine a cut-off value for afamin that could effectively predict the occurrence of GDM. The same researchers also point out the need to investigate the correlation between afamin levels and the occurrence of GDM in different ethnic groups [87].

In their scientific study, Atakul et al. [86] investigated afamin levels in women with GDM in the third trimester of pregnancy (n = 49) and compared them with healthy controls (n = 40). They also wanted to see whether afamin was suitable as a marker for the assessment of glycemic control. Moreover, in an attempt to better understand large for gestational age (LGA) fetal growth in women with GDM, they also checked whether afamin had a predictive value in relation to LGA fetal development. They found no correlation between afamin concentrations and the occurrence of GDM in pregnancy, however, they observed significantly increased afamin levels in GDM women with LGA fetuses compared to AGA fetuses, despite adequate glycemic control in the studied GDM women. However, they admit to a fundamental flaw in their study as they did not assess afamin levels in the studied patients in the first and second trimester of pregnancy, which made it impossible for them to compare the results from the third trimester to the results from the first and second trimester [86].

Eroğlu et al. [91] conducted a study in which they compared afamin and vitamin E levels in 43 women with GDM and 44 healthy controls. They observed that the mean concentrations of both molecules were higher in diabetic patients, but statistically this was not significant. They point out the need for further research to better understand the role of afamin and vitamin E in the pathogenesis of GDM [91].

Köninger et al. [90] evaluated pre-conceptional afamin and HOMA-IR levels and their predictive value in the occurrence of GDM in 63 women with PCOS. Twenty-nine of the study participants developed GDM during pregnancy (46%). Those patients who developed GDM during pregnancy had higher pre-conceptional values of afamin and HOMA-IR than women without GDM. According to the authors, a limitation of their study was that the time interval between the afamin assessments and the timing of conception differed among the study subjects. Nevertheless, the study results are suggestive of the potential usefulness of afamin assessment in predicting the development of GDM in women who are at an increased risk of developing the disease [90].

### 2.5. Adropin

Adropin is a novel regulatory protein encoded by the Energy Homeostasis (ENHO)-associated gene. It is mainly expressed in the liver and brain, but also in the kidneys, pancreas, and umbilical vein. Adropin is a regulatory factor in glucose and lipid homeostasis. It has been shown to correlate with obesity and is involved in the prevention of insulin resistance, dyslipidemia, and impaired glucose tolerance [94]. There is a noticeably higher level of adropin in patients with T2DM [95]. Considering the importance of adropin in energy homeostasis and insulin resistance, it is worth exploring the potential role of this protein in the pathogenesis of GDM. The available research into the changes in adropin levels in GDM is conflicting, with reports showing a significant increase or decrease in its levels in GDM patients compared with controls.

Beigi et al. [96] measured the serum adropin and lipid levels in women during the 24th–28th weeks of gestation with GDM and in healthy pregnant women who constituted a control group. The levels of adropin were significantly lower in the group of pregnant women with GDM. In addition, there was no significant correlation between the serum adropin levels and body mass index as well as fasting blood glucose levels or lipid profile. Beigi et al. [96] concluded that adropin is an independent predictor of GDM.

The study results of Dąbrowski et al. [97] showed that adropin levels were significantly higher in patients with GDM than in controls. Furthermore, in their study, a significant correlation was found between high adropin levels and elevated HbA1c levels. This may suggest a long duration of the disease and the possibility of carbohydrate disorders before pregnancy. Moreover, adropin and HbA1c have been found to be independent risk factors for endothelial dysfunction in patients with T2DM. In pregnancy, endothelial dysfunction may contribute to placental impairment in the first trimester and a higher incidence of late pregnancy complications. These include pre-eclampsia, impaired fetal growth, and intra-uterine fetal demise. Based on recent studies, it may be speculated that higher adropin levels in women with GDM are one of many adaptive responses to adverse glucose metabolism in pregnancy [97].

Vivek et al. [98], in their meta-analysis, also reported significantly elevated maternal serum adropin levels in patients with GDM compared with a control group without GDM. In addition, its serum levels in patients with late GDM were higher in the first and third trimesters of pregnancy. Adropin is known to be involved in the prevention of dyslipidemia, obesity, impaired glucose tolerance, and insulin resistance, which occur with increased frequency in patients with GDM [98]. Therefore, elevated adropin levels may be considered a protective compensatory mechanism that acts by suppressing hepatic glucose production and improving hepatic insulin sensitivity in patients with GDM.

### 2.6. Fetuin-A

Fetuin-A, also known as a2-HS-glycoprotein (AHSG), is a member of the cystatin protease inhibitor superfamily [99]. It is mainly synthesized and secreted by the liver and adipose tissue. Fetuin-A is involved in many physiological and pathophysiological processes in the human body [100]. Recent studies indicate that high levels of fetuin-A are associated with several metabolic disorders, such as insulin resistance, PCOS, and T2DM [100]. Fetuin-A is a ligand for TLR-4, through which lipids induce insulin resistance. In addition, high concentrations of AHSG induce inflammatory signaling [101]. Novel cross-sectional studies have shown a correlation between high levels of this glycoprotein and the risk of developing GDM.

Recent studies on fetuin-A levels in both normal and GDM-complicated pregnancies indicate significant associations between elevated serum AHSG levels and the development of GDM. The study results obtained by Iyidir et al. [102] revealed elevated serum fetuin-A levels in GDM patients in comparison to healthy controls. In addition, it was observed that concentrations decreased during the postpartum period. The same researchers also revealed a correlation between the levels of fetuin-A and HbA1c [102].

Šimják et al. [103] found elevated serum levels of fetuin-A both in women who had normal pregnancies and in those whose pregnancies were GDM-complicated. After delivery, fetuin-A levels were decreased only in the group of mothers with a history of uncomplicated pregnancies. In contrast, women with GDM had similar levels of fetuin-A before and after delivery [103].

The study presented by Jin et al. [104] describes the changes in fetuin-A levels depending on the gestational age of GDM pregnancies. The authors showed that in both the first and second trimesters, the fetuin-A levels were higher in women with GDM compared to healthy controls. A significant increase in fetuin-A levels occurred between the first and second trimesters and it was an independent risk factor for GDM. This fact emphasizes the importance of monitoring the dynamic changes in the fetuin-A concentrations during pregnancy. The increase in fetuin-A levels may be related to disturbances in insulin sensitivity and the function of the β-cells [104].

A topic worth elaborating is the concentration of fetuin-A in the cord blood in pregnancies with GDM compared with euglycemic pregnancies and its effect on fetal development. Wang et al. [105] observed a negative correlation between fetuin-A and birth weight and birth length only in the GDM group. Healthy controls displayed no such association, suggesting that under normal physiological conditions, fetuin-A may have no effect on fetal growth.

### 2.7. Zonulin

Zonulin is a physiological modulator of intercellular tight junctions (TJs) between the intestinal epithelial cells. Gliadin and bacteria induce the secretion of zonulin mainly from the liver [106]. This leads to a loss of protein interaction, resulting in increased intestinal permeability, introducing foreign antigens into the immune system, and causing inflammation [107]. Increased zonulin levels are observed in autoimmune diseases associated with TJ dysfunction, including celiac disease [108,109]. In addition, zonulin has been implicated in the pathogenesis of neurodegenerative diseases and cancer. Recent studies investigating the association between increased serum zonulin levels and the probability of developing GDM are very promising.

One of the first studies examining zonulin as a potential marker for GDM was conducted by Mokkala et al. [110]. They demonstrated that its serum levels were higher in women diagnosed with GDM in comparison to the control group. Importantly, zonulin concentration measured in early pregnancy, before the diagnosis of GDM, was higher. Thus, the measurement of zonulin concentration can be taken into account as a predictor of an increased risk of GDM [110].

A result consistent with the above reports was obtained by Bawah et al. [111]. They determined the serum zonulin levels in pregnant women between 11 and 13 weeks of gestation. Their study showed that significantly higher serum zonulin levels were detected in the women who later developed GDM compared to those who remained normoglycemic throughout pregnancy [111]. Higher serum levels in women with GDM were also demonstrated by Demir et al. [112] Moreover, they showed that the plasma zonulin levels positively correlated with BMI, fasting plasma glucose, HbA1c, and HOMA-IR in GDM [112].

Considering the previous studies, it seems that zonulin can be used as a predictive, non-invasive biomarker in the diagnosis of GDM. Its increased levels do not merely affect increased intestinal permeability, but they may also reflect a response secondary to inflammation and insulin resistance in GDM. Presumably, zonulin is involved in the development of GDM by interfering with insulin receptor function and stimulating inflammation.

### 2.8. SFRPs

SFRPs produced by adipose tissue are members of soluble, extracellular signaling ligands with a cysteine rich domain. SFRPs are the Wingless-related integration site (Wnt) signaling pathway antagonistic inhibitors that act through binding with the extracellular Wnt-5a or Wnt-3a [113,114,115]. The Wnt pathway is responsible for organismal growth and differentiation. In humans, five soluble, secreted glycoproteins, such as SFRP1, SFRP2, SFRP3, SFRP4, and SFRP5, are distinguished and coded by the gene on chromosome 10 [116]. The largest member of the SFRP family is SFRP4. SFRPs are also described as adipokines, which take part in adipogenesis [117,118,119]. They are expressed in many tissues, such as the urinary bladder, bone marrow, spleen, pancreas, and liver, and they take part in adipocyte differentiation, pathogenesis of hypophosphatemic diseases, bone cancers, CVDs, retinal degeneration, diabetes mellitus, and others [120,121,122]. SFRPs may be detected in serum and urine.

The molecular mechanisms of SFRP1, SFRP4, and SFRP5 may have a direct or indirect effect on the development of diabetes. The overexpression of SFRP1 may worsen insulin secretion by acting on the β cells through β-catenin, transcription factor 4 (TCF4), and CyclinD [123]. In contrast to SFRP1, SFRP3 is negatively related to the occurrence of diabetes. The ongoing inflammatory process and its components, such as IL-6 and interferons, cause a decrease in the level of SFRP3, which is responsible for sensitizing the skeletal muscle cells to insulin.

SFRP4 via the Wnt/Ca^2+^ signaling pathway enhances the intracellular Ca^2+^ and protein kinase C. Due to these interactions, calmodulin kinase II is activated. The overexpression of SFRP4 is positively and indirectly related to the development of diabetes. Some researchers report a positive correlation between SFRP4 and the level of HbA1c and fasting triglyceride [122,124,125,126]. In addition, glucose intolerance may occur due to the IL-1β stimulation to release more SFRP4 in serum [127]. The expression of SFRP4 is also connected with miR-30d, miR-146a, and miR-24, which are elevated in the serum of patients with diabetes [128]. SFRP4 leads to oxidative stress in the pancreas and impairs the exocytosis of insulin [129].

SFRP5 negatively interacts with the insulin receptor substrate-1 (IRS-1) [130,131,132]. Insulin resistance, which develops in pregnancy, is connected with increasing BMI values [133,134]. A study conducted by Trojnar et al. revealed a positive correlation between the level of SFRP5 and pre-pregnancy BMI, BMI at delivery, total cholesterol, and LDL, as well as the levels of triglycerides in women suffering from GDM [135]. The opposite results were observed in healthy controls [135].

Oztas et al. also proposed that the increased levels of SFRP5 may contribute to the development of GDM [136]. Hu et al. [137], Lu et al. [138], and Toan et al. [139] showed that the level of SFRP5 is similar in women who suffer from diabetes and/or are obese. SFRP5, which is an antagonist for the Wnt/β-catenin signaling pathway, may steer the hypothalamic insulin signaling pathway, and it also inhibits the N-methyl-D-aspartate receptor and the secretion of hepatic glucose [140]. In another scientific study, Christodoulides et al. demonstrated that SFRP5 may prompt the epigenetic silencing of the Wnt signaling pathway in white adipose tissue, which could lead to increased adipogenesis [141]. Furthermore, SFRP5 inhibits the activation of the c-JunN-terminal kinase (JNK) downstream of the Wnt signaling pathway. Mori et al. pointed out that SFRP5 promotes adipocyte growth by worsening oxidative metabolism [142]. Further studies conducted by Rulifson et al. [143] and Van Camp et al. [144] confirmed the thesis presented above. Elevated SFRP5 levels were detected in obese patients (basing on BMI, waist–hip ratio, body fat percentage, and lipid profile) who developed GDM. Prats-Puig et al. noticed that SFRP5 may be an adipokine that is negatively regulated during advancing obesity, which can ultimately lead to an abnormal metabolic phenotype [145]. It is worth noting that maternal weight may have an impact on metabolic imbalance in the newborn. The umbilical cord SFRP5 levels may positively correlate with the maternal concentrations of total cholesterol and HbA1c. Kimber-Trojnar et al. [146] found a positive association between the umbilical cord SFRP5 and leptin. Moreover, there was a negative correlation between the SFRP5 and ghrelin concentrations in the umbilical cord of GDM mothers [146].

The role of SFRPs in diabetes is still awaiting clarification. There are limited data concerning SFRPs in obstetrics. A number of signaling pathways have been reported to be implicated in the β-cell failure. Therefore, further studies are necessary to assess the prognostic value of the presented biomolecules in monitoring patients who are in the risk group.

### 2.9. Amylin

Amylin (i.e., Islet Amyloid Pancreatic Polypeptide; IAPP) is a hormone secreted along with insulin by pancreatic β-cells in a pulsatile pattern and seems to play a significant role in the regulation of glucose metabolism [147]. It helps to control gastric emptying, suppression of glucagon release, and regulation of satiety [148,149]. Amylin is the major component of amyloid, which is detected in the islet of Langerhans present in more than 90% of patients with T2DM. Amylin mature fibrils induce β-cell cytotoxicity through their penetration of the cell membrane, resulting in an imbalance of intracellular ions [150,151]. This results in the formation of reactive oxidant species, membrane damage, and the loss of β-cells, which leads to T2DM development [152].

Pregnancy is a condition with reduced tissue sensitivity to insulin, so in order to remain normoglycemic, it is necessary to increase insulin secretion. This effect is achieved by increasing β-cell mass. When this is not reachable, there is a risk of developing GDM [152]. It is possible that there is a similar mechanism of pathogenesis as in T2DM—amylin cytotoxicity leads to β-cell impairment and apoptosis, resulting in decreased insulin secretion and reduced glucose tolerance.

Grigorakis et al. [153] noted that the expression of amylin was not found in the placentas obtained from 5 women with GDM and 5 healthy mothers. However, the serum levels of amylin relative to insulin were decreased in the GDM group [153]. Different results were obtained in the study by Wareham et al., in which gestational diabetes mellitus was associated with the increased serum concentration of amylin [154].

Kinalski et al. [155] revealed that levels of amylin were increased in mothers with GDM in the puerperium in comparison to healthy women.

Although there is a potential link between the effect of amylin cytotoxicity and the development of GDM, evidence is lacking and there is a need for further research to confirm this association.

## 3. Conclusions

In this day and age, searching for predictive molecules capable of detecting a disease even before its diagnosis and occurrence of complications, which is more often than not irreversible, is going on in every field of medicine. Scientists are still in search of the most appropriate and rapid diagnostic strategy, involving novel biomarkers to strengthen the screening and improve diagnostic processes.

The period of pregnancy is said to be “a window to future health”, therefore the occurrence of characteristic complications during pregnancy may trigger a metabolic or vascular risk of developing civilization diseases in the future life of the mother. Gestational changes, including both the development of insulin resistance and an increase in body fat, can be “a double-edged sword”. Weight gain and metabolic disturbances during pregnancy, if left unattended to, may have negative consequences for women, thereby increasing the risk of long-term complications. It is well known that GDM is the “window” to the diagnosis of CVDs, T2DM, obesity, and metabolic syndrome in the mother later in her life. On the other hand, GDM also has a huge impact on fetal programming. The intra-uterine development of the fetus of the mother suffering from GDM “programs” towards civilization diseases that may appear later in the life of her offspring.

This merits widespread attention since it is extremely important that pregnant women promote long-term health, including healthy eating habits, having more physical exercise, and regular check-ups. Ideally, the course of pregnancy should be free of all pathologies, without the need for hospitalization and treatment. It is worth emphasizing that many medications taken in pregnancy may be harmful for the growing fetus. In GDM, if the diet is insufficient, the available treatment options include the administration of insulin or oral antidiabetic medicaments. In obstetrics, one should think twice before any decision about treatment is made. Therefore, finding early predictors capable of signaling any risk of developing a certain disease, e.g., GDM, in the first trimester of pregnancy is of the utmost importance.

For many years, scientists have been searching for molecules in the serum or biological material, i.e., urine or saliva, to perform thorough laboratory determinations of a pregnant woman’s condition.

In this review, which is a follow-up to the manuscript published in October 2021 [40], we present some more, less known biomolecules, i.e., CMPF, ANGPTL-8, nesfatin-1, afamin, adropin, fetuin-A, zonulin, SFRPs, and amylin, which could extend the list of early predictors and help to better plan the strategy of maternity care, thereby reducing the risk of negative consequences and complications caused by GDM.

Our study investigates the 9 biomolecules with different mechanisms of action (Table 1). Moreover, 7 of them present controversial data concerning the serum concentrations in women with GDM compared to healthy pregnant women (Figure 1).

In light of the presented studies, it seems that high levels of fetuin-A and zonulin in the serum of pregnant women can be used as predictive markers in the diagnosis of GDM. We do hope that further clinical research will shed more light on the presented hypotheses, and, what is of the utmost importance, identify the biomolecule with the greatest potential to predict the development of GDM development.

## Figures and Tables

**Figure 1 ijms-23-04364-f001:**
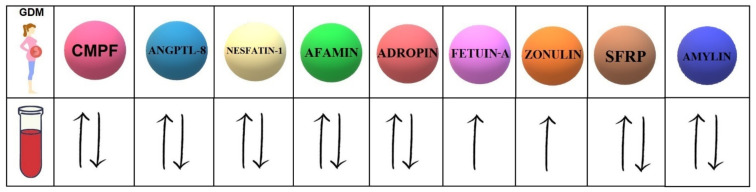
Concentrations of selected biomolecules in the serum of GDM patients compared to their concentrations in the serum of healthy pregnant women. CMPF (3-carboxy-4-methyl-5-propyl-2-furanpropanoic acid); ANGPTL-8 (angiopoietin-like protein-8); SFRPs (secreted frizzled-related proteins).

**Table 1 ijms-23-04364-t001:** Potential mechanisms of action of selected biomolecules.

Biomolecules	Localization	Mechanism of the Action
**CMPF**	Liver, pancreas	stimulates oxidative stress, acts on mitochondrial functions and dysregulates glucose-induced ATP accumulation [49]
**ANGPTL-8**	Adipose tissue, liver, vascular and hematopoietic systems	supporting role in stimulating proliferation and increasing pancreatic beta cell mass, improves glucose tolerance in insulin resistance, likely by increasing insulin secretion [64,67],
**Nesfatin-1**	Pancreas, duodenum, adipose tissue, peripheral and central nervous system (arcuate, paraventricular nuclei and nucleus of the solitary tract), skeletal muscles, heart, kidneys, liver, skin, lungs, articular cartilage	stimulates pre-proinsulin mRNA expression and increases glucose-induced insulin release [78], inhibits glucagon secretion [78], antihyperglycemic effects in animal studies [81]
**Afamin**	Liver, placenta, ovarian follicular, seminal fluids, cerebrospinal fluids, plasma	antiapoptotic and antioxidant qualities [90,91], upregulation of oxidative stress increases concentrations of afamin [86,90]
**Adropin**	Lungs, liver, cardiovascular system, adipose tissue, kidneys, pancreas, brain, cerebellum, small intestine, endothelial cells	regulates cellular energy metabolism and anti-inflammatory processes (IL-10↑, TGFβ↑, IL-12↓, TNFα↓), takes part in anti-oxidative stress [95,96]
**Fetuin-A**	Skeletal muscles, adipose tissue, cardiovascular system, liver	a ligand for toll-like receptor 4 (TLR-4), through which lipids induce insulin resistance, induce inflammatory signaling [101]
**Zonulin**	Digestive system, liver, heart, brain, adipose tissue, lungs, kidneys, skin, immune cells	physiological modulator of intercellular tight junctions (TJs) between intestinal epithelial cells [106]
**SFRPs**	Heart, adipose tissue, pancreas, skeletal muscles, liver, aorta, endometrium, gallbladder, kidneys, prostate, testis, urinary bladder, ovary, esophagus, skin, small intestine, colon, appendix, spleen, bone marrow, duodenum, adrenal	extracellular signaling ligands and Wnt signaling pathway antagonistic inhibitors [113,114,115], take part in the adipogenesis [117,118,119], adipocyte differentiation [120,121,122], may worsen insulin secretion by acting on islet cells through β-catenin, TCF4, CyclinD [123], sensitizing skeletal muscle cells to insulin. By the Wnt/Ca^2+^ signaling pathway, enhance intracellular Ca^2+^ and protein kinase C, calmodulin kinase II is activated [122,124,125,126], steer hypothalamic insulin signaling pathway, inhibits N-methyl-D-aspartate receptor and inhibits the secretion of hepatic glucose [140]
**Amylin**	Pancreatic β-cells	plays a significant role in regulation of glucose metabolism [147], controls gastric emptying, suppression of glucagon release and regulation of satiety [148,149], penetrates cell membranes [152], resulting in an imbalance of intracellular ions, formation of reactive oxidant species, membrane damage and loss of β-cells [150,151]

## Data Availability

Not applicable.

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
