# Peer review of "Novel Biomolecules in the Pathogenesis of Gestational Diabetes Mellitus 2.0"

_ijms, 2022, doi:10.3390/ijms23084364_

Round 1

Reviewer 1 Report

This review summarized some of the novel biomolecules associated with the pathogenesis of gestational diabetes mellitus, which was added as a complement to a similar review they recently published. The discovery of biomarkers is important for the early diagnosis of the onset of  GDM. The topic is interesting. Overall, this manuscript was well written and organized. However, there are some concerns that are needed to be addressed.

  1. Some descriptions are redundant or repetitive in the text, such as many sentences in the last paragraph of the Introduction were similar to those in the Abstract. 
  2. The conclusion is needed to be condensed. Many descriptions are redundant.
  3. Islet amyloid polypeptide has been reported to be associated with the pathogenesis of GDM although there are controversies. The authors should briefly review these studies.

Author Response

Dear Reviewer 1,

            We would like to resubmit our manuscript entitled “Novel biomolecules in the pathogenesis of gestational diabetes mellitus 2.0”. We appreciate your valuable remarks and hope that the quality of our manuscript is going to meet your expectations now that we have made some suggested alternations.

 “This review summarized some of the novel biomolecules associated with the pathogenesis of gestational diabetes mellitus, which was added as a complement to a similar review they recently published. The discovery of biomarkers is important for the early diagnosis of the onset of  GDM. The topic is interesting. Overall, this manuscript was well written and organized. However, there are some concerns that are needed to be addressed.”

Thank you very much for finding the time to read our manuscript. Thank you for considering our manuscript. We find all your remarks spot on therefore we have made a point-by-point correction of the manuscript according to your suggestions.

“Some descriptions are redundant or repetitive in the text, such as many sentences in the last paragraph of the Introduction were similar to those in the Abstract.”

Following your advice, we have rephrased the Introduction.

“The conclusion is needed to be condensed. Many descriptions are redundant.”

Following your advice, we have reedited the part entitled “ Conclusions”.

“Islet amyloid polypeptide has been reported to be associated with the pathogenesis of GDM although there are controversies. The authors should briefly review these studies.”

Following your advice, we have introduced the part entitled “2.9. Amylin” and additional references.

            We would like to take this opportunity to thank you for all the valuable and highly perceptive remarks which have definitely made a substantial contribution to the quality of our paper.

Yours faithfully,

Prof. Zaneta Kimber-Trojnar

Chair and Department of Obstetrics and Perinatology

Medical University of Lublin, 20-090 Lublin, Poland

Tel: +48-81-7244-769

Reviewer 2 Report

The presented article is aimed to review the potential and importance of eight, less known, molecules in the pathogenesis of Gestational diabetes mellitus. I feel that the manuscript is well written and structured and provides relevant and interesting new information on the subject.

My coments can be found below:

The abstract should contain a summary of the topic, the methodology and the main conclusions of the review. No data from previous studies should be reported.

Introduction: the first 2 lines are the same as the ones in the abstract. Please rewrite it.

Conclusion need to be rewrite: Authors should not mention any example in conclusions. This part is to describe and highlight conclusions after the general revision of the topic revised.   You should include main findings and emphasise the importance of what you want to evidence, avoiding the use of general statements providing filler information. 

Author Response

Dear Reviewer 2,

            We would like to resubmit our manuscript entitled “Novel biomolecules in the pathogenesis of gestational diabetes mellitus 2.0”. We appreciate your valuable remarks and hope that the quality of our manuscript is going to meet your expectations now that we have made some suggested alternations.

“The presented article is aimed to review the potential and importance of eight, less known, molecules in the pathogenesis of Gestational diabetes mellitus. I feel that the manuscript is well written and structured and provides relevant and interesting new information on the subject.”

Thank you very much for finding the time to read our manuscript. Thank you for considering our manuscript. We find all your remarks spot on therefore we have made a point-by-point correction of the manuscript according to your suggestions.

“The abstract should contain a summary of the topic, the methodology and the main conclusions of the review. No data from previous studies should be reported.”

Following your advice, we have rephrased the article abstract.

“Introduction: the first 2 lines are the same as the ones in the abstract. Please rewrite it.”

Following your advice, we have reedited the Introduction.

“Conclusion need to be rewrite: Authors should not mention any example in conclusions. This part is to describe and highlight conclusions after the general revision of the topic revised. You should include main findings and emphasise the importance of what you want to evidence, avoiding the use of general statements providing filler information.”

Following your advice, we have rephrased the part entitled “ Conclusions”.

Yours faithfully,

Prof. Zaneta Kimber-Trojnar

Chair and Department of Obstetrics and Perinatology

Medical University of Lublin, 20-090 Lublin, Poland

Tel: +48-81-7244-769
